# A Voting-Based Ensemble Deep Learning Method Focused on Multi-Step Prediction of Food Safety Risk Levels: Applications in Hazard Analysis of Heavy Metals in Grain Processing Products

**DOI:** 10.3390/foods11060823

**Published:** 2022-03-13

**Authors:** Zuzheng Wang, Zhixiang Wu, Minke Zou, Xin Wen, Zheng Wang, Yuanzhang Li, Qingchuan Zhang

**Affiliations:** 1School of Economics & Management, Nanjing Tech University, Nanjing 211816, China; zzwang@njtech.edu.cn (Z.W.); xwen@njtech.edu.cn (X.W.); 2School of Physical and Mathematical Sciences, Nanjing Tech University, Nanjing 211816, China; mkzou@njtech.edu.cn; 3National Engineering Laboratory for Agri-Product Quality Traceability, Beijing Technology and Business University, Beijing 100083, China; wzbtbu@163.com; 4School of Computer Science and Technology, Nanjing Tech University, Nanjing 211816, China; yzli@njtech.edu.cn

**Keywords:** food safety risk assessment, risk level classification, grain processing products, heavy metal hazard, multi-step time series prediction, deep learning

## Abstract

Grain processing products constitute an essential component of the human diet and are among the main sources of heavy metal intake. Therefore, a systematic assessment of risk factors and early-warning systems are vital to control heavy metal hazards in grain processing products. In this study, we established a risk assessment model to systematically analyze heavy metal hazards and combined the model with the K-means++ algorithm to perform risk level classification. We then employed deep learning models to conduct a multi-step prediction of risk levels, providing an early warning of food safety risks. By introducing a voting-ensemble technique, the accuracy of the prediction model was improved. The results indicated that the proposed model was superior to other models, exhibiting the overall accuracy of 90.47% in the 7-day prediction and thus satisfying the basic requirement of the food supervision department. This study provides a novel early-warning model for the systematic assessment of the risk level and further allows the development of targeted regulatory strategies to improve supervision efficiency.

## 1. Introduction

Food quality and safety issues have drawn wide interest with the continuous development of the social economy [1]. Governments worldwide have exerted considerable efforts to improve food safety [2]. In China, more stringent regulatory measures have been implemented by the government. Despite these efforts, food safety incidents still arise [3]. Food safety concerns challenge the food safety oversight system of the country and pose an economic threat [4]. One reason is that most supervision measures and methods rely on manpower, and a severe shortage of qualified professional supervision talents is a current concern [5]. Meanwhile, heavy metal deposition in agroecosystems has recently increased, particularly in northern China, posing serious risks to crop safety and human health via the food chain [6]. The quality and safety of grain processing products such as wheat and its products, which are vital food and feed ingredients in China, have gained interest [7]. Meanwhile, grain processing products have become the main source of heavy metal intake among residents in China [8]. To reduce the likelihood of heavy metal contamination at a reduced cost, big data and artificial intelligence methods need to be applied for efficient monitoring of safety issues. Moreover, appropriate food safety assessment methods have to be implemented to determine the effect of heavy metal contamination on the safety of grain processing products.

The actual situation in China indicates that heavy metals are likely to increase non-carcinogenic and carcinogenic health risks [9]. For instance, exposure to cadmium has been associated with numerous adverse health effects, including atherosclerosis, cancer, and possibly melanoma [10]; we established a dietary exposure assessment model to analyze the food safety risks of heavy metals attributed to environmental factors, carcinogenic, and non-carcinogenic health factors in grain processing products. A comprehensive risk assessment and a hazard analysis of heavy metals were conducted using the analytic hierarchy process based on entropy weight (AHP-EW), combined with K-means++ clustering. Accordingly, we proposed an improved early-warning model based on a voting-ensemble method, which integrates deep learning models in the multi-step prediction. The effectiveness of the proposed early-warning approach was validated by grain processing product detection data from the National Food Safety Sampling Inspection Information System and then the proposed model was compared with current models. This approach benefits food safety supervision departments by reducing manpower supervision costs and can effectively predict food safety risks.

## 2. Background Studies

### 2.1. Food Safety Risk Assessment and Classification

Food quality and safety are closely related to the health and living standard of individuals, and the risk assessment of food quality and safety bears considerable social significance [11]. A food safety risk assessment and early-warning analysis have recently been conducted. Several studies on risk assessment [11,12,13] have applied AHP-EW to determine objective food safety risk values as inputs in early-warning models. However, studies use single risk values as the assessment index and thus lack a comprehensive risk assessment. A food safety assessment has to consider the effect of food pollutant exposure on human beings. Accordingly, B. Niu et al. [14] established dietary exposure models, which are typically used to assess the carcinogenic and non-carcinogenic risks of children and adults after metal exposure [15], allowing for a comprehensive assessment of the health risks in vegetables and providing scientific and comprehensive support for risk assessments.

For a comprehensive assessment of food safety risks, many dietary exposure assessment models have been explored, including the target hazard quotient (THQ) and target cancer risk (TCR) established by the United States Environmental Protection Agency (US EPA) in 2000. THQ is the pollutant exposure dose and reference dose to characterize the non-carcinogenic risk of pollutants [16]. TCR is based on the pollutant exposure dose and carcinogenic intensity index. The index reflects the possible type of carcinogenic risk [17]. Moreover, the Nemerow integrated pollution index (NIPI) is a water pollution index used to evaluate heavy metal pollution in soil or sediment [18]. Considering the need to integrate the heavy metal hazards of grain processing products with environmental and health factors, we introduced food safety risk assessment indexes—TCR, THQ, and NIPI—to comprehensively measure the heavy metal hazard in grain processing products and used them as inputs in early-warning models.

With regard to risk classification, Geng et al. and Ma et al. used the interval distribution or risk matrix of the risk value, respectively [13,19], to establish food safety risk levels. However, risk level classification based on risk values, rather than assessment values, fails to comprehensively reflect the risks associated with heavy metals, and risk level classification based on the interval distribution or risk matrix is subjective [19]. As a machine learning method, the clustering algorithm classifies samples based on sample similarity in a data-driven manner [20,21]. Thus, the influence of subjective factors is effectively reduced, and index prediction is converted to level prediction. Therefore, we decided to combine risk assessment indexes with K-means++ clustering (an improved clustering algorithm) to realize a comprehensive assessment and objective classification of heavy metal hazards in grain processing products.

### 2.2. Early-Warning Models of Food Safety Risk

In early research, common early-warning models of food safety risk, including models based on a grey relational analysis (GRA) [22] and artificial neural networks (ANNs) [23], were applied to food safety prediction problems. Han et al. [11] developed multiple GRA models to forecast food quality and safety. Lin et al. [24] adopted a GRA model that integrates interpretative structural modeling to analyze the factors influencing food safety.

In relatively recent studies, neural networks were applied in food safety early-warning models. Geng et al. [25] used the radial basis function (RBF) as the element to construct a deep RBF model for early-warning modeling of complex food detection data. Geng et al. also used the agglomerative hierarchical clustering radial basis function (AHC-RBF) neural network to adaptively obtain the central position of the hidden layer nodes of the RBF, thus improving the prediction precision of the RBF [13].

However, typical shallow neural network methods, such as ANN, back propagation (BP), and RBF, may not be able to extract and use deep features [26]. However, deep learning methods such as long short-term memory (LSTM), gated recurrent units (GRUs), and recurrent neural networks (RNNs) can suitably capture high-dimensional features and exhibit temporal dynamic behavior [27]. These approaches have been employed in weather forecasting or travel-time prediction, achieving enhanced accuracy [28]. Meanwhile, the voting ensemble method can overcome limitations to several algorithms [29] and decrease the variance in a trained model on the validation set [27].

The majority of current prediction methods used in food safety are single-step prediction or fitting prediction techniques, which cannot predict unknown data, that is, future occurrences. As a significant research area in data analysis, time series forecasting plays an important role in the processing industry [30], clinical medicine [31], and other sectors [32] because of its capability to analyze the historical data of a dynamic system and predict future operation patterns [33]. This feature is consistent with the requirement of food safety risk prediction. Therefore, a multi-step time series prediction is more valuable than a single-step prediction [34], and the same is true for food safety. Considering the actual requirement of the food supervision department, we proposed a voting-ensemble technique that integrates deep learning models to grasp the long-term change trend of food safety risks, realizing a more accurate multi-step prediction of food safety risk than that of shallow NNs.

The specific food safety risk assessment and early-warning model proposed in this study is presented in Figure 1.

As shown in Figure 1, the newly proposed method of classifying and predicting food safety risk levels, integrated with assessment indexes, mainly consists of three blocks. In the assessment blocks, the dietary assessment of heavy metal hazards is conducted. In the classification block, a clustering algorithm is employed to determine the risk level. In the prediction block, we applied a voting-based ensemble deep learning method to implement the multi-step prediction.

## 3. Materials and Methods

### 3.1. Materials

#### 3.1.1. Data Sources

In this study, three heavy metals that cause heavy metal pollution in grain processing products are selected as the research objects: chromium [35], cadmium [14], and arsenic [36]. A total of 65,302 samples from the 2020 National Food Safety Sampling Inspection Information System are included: chromium (12,501), cadmium (29,456), and arsenic (23,795). Descriptions of several detection data are shown in Table 1.

The detection data cover the 20 provinces of China from March to December 2020 and are characterized by high-dimensional attributes, complexity, discreteness, and nonlinearity, which are reflected in the distribution of the mean values of the three heavy metals’ daily detection data in Figure 2 and the description of several observations in Table 1 [12].

To establish the subsequent risk assessment model, we collect the resident consumption data and related toxicology data to calculate the assessment indexes. The data on the resident consumption of grain processing products in the 20 provinces shown in the Table 2 come from the Fifth Chinese Total Diet Study [8], which adopts stratification based on population proportions and multi-stage cluster random sampling to conduct a dietary questionnaire survey on the main foods consumed by residents.

Moreover, related toxicology data are acquired from reports or bibliographic searches of international organizations, such as the Food and Agriculture Organization of the United Nations, the World Health Organization (WHO) Joint Expert Committee on Food Additives, and the United States EPA. The reference doses of chromium, cadmium, and arsenic are 0.003 (trivalent chromium) μg/(kg d), 0.001 μg/(kg d), and 0.0003 μg/(kg d), respectively [37]. The cancer slope factor (CSF) of chromium, cadmium, and arsenic are 0.5 (kg d)/mg, 6.3 (kg d)/mg, and 1.5 (kg d)/mg, respectively [38].

#### 3.1.2. Data Preprocessing

Some detection results are recorded as “not detected” in the original data. With the requirement to predict the levels of food safety risk considered, these results are replaced by half of the metal detection line instead of being directly replaced with zero [39] in accordance with the principle of the credible evaluation of low pollutant levels proposed at the second meeting of the WHO Global Environment Monitoring System/Food [14]. For results with an extra symbol such as “<” the symbol is deleted, and the value is retained [12]. Moreover, the detection results for total arsenic in food are converted to inorganic arsenic at a ratio of 70% to calculate the exposure amount [40].

### 3.2. Food Safety Risk Assessment and Classification

#### 3.2.1. Assessment Indexes

To improve the accuracy of predicting the food safety risk level and measure the precise effect of heavy metal hazards in grain processing products on the human body, the following safety indexes are selected in this study to classify the daily risk levels.

The NIPI, which reflects the characteristics of food pollution, is used to evaluate heavy metal pollution in rice [16], air [41], and water [16]. The NIPI $P_{c\left(i,j\right)}$ of the heavy metal j in grain processing products i is given by
(1)$$P_{c\left(i,j\right)}=\sqrt{\frac{P_{\max\left(i,j\right)}^{2}+P_{avg\left(i,j\right)}^{2}}{2}}$$
where $P_{\max\left(i,j\right)}$ is the maximum value of the heavy metal j pollution index in grain processing products, and $P_{avg\left(i,j\right)}$ is the average value. The specific expression of the pollution index is expressed as
(2)$$P_{i,j}=\frac{X_{i,j}}{S_{i,j}}$$
where $P_{i,j}$, $X_{i,j}$ are the pollution index and detection value of heavy metal j in food i, respectively, and $S_{i,j}$ is the national limit standard for heavy metal j in food i. In this study, food i denotes the grain processing, and j represents chromium, cadmium, and arsenic. The detection values of the grain processing products with the same data are substituted into Equations (1) and (2) to calculate the NIPI of the three heavy metals on a certain day.

Considering the carcinogenicity of heavy metals, we use the TCR to measure the carcinogenic risk. Meanwhile, the non-carcinogenic risk is given by THQ, which is based on the pollutant exposure dose and reference dose [42]. The TCR is based on pollutant exposure dose, and the carcinogenic intensity index of the product reflects the possible type of carcinogenic risk [43].

The specific expression of TCR is given by
(3)$$TCR=\frac{EF\times ED\times CSF_{j}\times EDI_{j}^{50}}{AT_{C}}$$
where EF is the exposure frequency (365 days/year); ED is the exposure period (70 years in the current study); $CSF_{j}$ denotes the carcinogenic intensity index of the heavy metal j (kg·d/mg); $AT_{C}$ is the duration of the carcinogenic effect (365 days/years*exposure period, assumed to be 70 years in this study). $EDI_{j}^{50}$ is calculated using
(4)$$EDI_{j}^{50}=\frac{FC_{j}\times X_{i,j}^{50}}{W}$$
where $FC_{j}$ is the per capita daily consumption of China’s grain processing products i (kg/d); $X_{i,j}^{50}$ is the 50th quantile (mg/kg) of the heavy metal j detected on a certain day; and W is the average body mass of the residents (60 kg in this study).

Similarly, THQ is expressed as
(5)$$THQ=\frac{EF\times ED\times EDI_{j}^{95}}{AT_{C}\times RfD_{j}}$$
where $RfD_{j}$ (reference dose) is the oral reference dose of the heavy metal j (kg·d/mg). $EDI_{95}$ is calculated as
(6)$$EDI_{j}^{95}=\frac{FC_{j}\times X_{i,j}^{95}}{W}$$
where $X_{i,j}^{95}$ is the 95th quantile (mg/kg) of heavy metal j detected on a certain day.

#### 3.2.2. Analytic Hierarchy Process Base on Entropy Weight

To reduce the scale of data and comprehensively measure the risk of heavy metal contamination in grain processing products in China on a certain day, this study determines the comprehensive assessment indexes by using the AHP-EW [12]. The assessment indexes of the three heavy metals are combined using the weight vector $W={\left[w_{1},w_{2},\ldots ,w_{n}\right]}^{T}$. The fusion data point Y is calculated using Equation (7)
(7)$$Y=X^{T}W$$
where X is the value matrix of the assessment indexes.

The food safety risk assessment index is calculated based on AHP-EW (Figure 3).

Therefore, on the basis of the AHP-EW method, this study integrates the indexes NIPI, TCR, and THQ of the three heavy metals into comprehensive NIPI, comprehensive TCR, and comprehensive THQ, namely, the food safety risk assessment index. These indexes are then used as the basis of risk classification and the input vector of the subsequent prediction model.

#### 3.2.3. Clustering Risk Classification

The K-means++ clustering algorithm, which exhibits low complexity, rapid computational capability, the ability to handle large data sets, and the flexibility to adjust the cluster number [44], is used to determine the risk level of a heavy metal hazard on the basis of assessment indexes. The specific process of K-means++ is as follows [45]:

1.(a) Take one center $\mu_{1}$ as the initial cluster center, chosen uniformly from the samples.1.(b) For each sample $x_{i}$, calculate $d\left(x_{i}\right)$, i.e., the shortest distance between sample $x_{i}$ to the closest center which has already been selected.1.(c) Choose one of the samples as the new cluster center $\mu_{1}$ according to the weighted probability:



(8)
$$M=\frac{d{\left(x_{i}\right)}^{2}}{\sum_{i=1}^{n}d{\left(x_{i}\right)}^{2}}$$



1.(d) Repeat steps 1 (b) and 1 (c) until cluster centers n have been chosen.2.Update the labels $y_{1},y_{2},\ldots ,y_{n}$ which correspond to the samples $x_{1},x_{2},\ldots ,x_{n}$:


(9)
$$y_{i}\leftarrow \underset{y\in \{1,2,\ldots ,n\}}{\arg\;\min}{\left\|x_{i}-\mu_{y}\right\|}^{2},i=1,2,\ldots ,n$$


3.Update a new center for each cluster $\mu_{1},\mu_{2},\ldots ,\mu_{n}$:


(10)
$$\mu_{y}\leftarrow \frac{1}{n_{y}}\underset{i:y_{i}=y}{\Sigma }x_{i},y=1,2,\ldots ,n$$


where $n_{y}$ denotes the number of samples belonging to label y.

4.Repeat step 2 and step 3 until the convergence has been reached.

Alternatively, by calculating the silhouette coefficient, which takes both cohesion and separation into account, one can determine the best cluster number [46]. The silhouette coefficient can be expressed as follows [47]:(11)$$s_{j}=\frac{b_{j}-a_{j}}{\max\left(a_{j},b_{j}\right)}$$
where, $a_{i}$ is the average distance of the jth sample to all other samples in the same cluster; $b_{i}$ is the average distance of the jth sample to all other provinces in different clusters. Then, the optimal cluster number can be obtained by calculating the average silhouette coefficient of all the samples. For one clustering with k categories, the average silhouette coefficient refers to the average of silhouette coefficients of samples belonging to the cluster and given as follows:(12)$$SC_{k}=\frac{1}{n}\sum_{j=1}^{n}s_{j}$$
where, n is the total number of samples in the data set. Besides, a higher value represents better clustering quality. Thus, the optimal clustering results are obtained. With regard to the risk threshold, the risk level based on a clustering algorithm is calculated as follows:(13)$$d_{level}=\min\left\{d_{1},d_{2},\cdots ,d_{n}\right\}$$
where $d_{k},k=1,2,\cdots ,n$ represents the distance between the jth sample and the center of the kth class of the clusters; $d_{level}$ denotes the minimum value of the distance between that jth sample and each cluster center. If the $d_{level}$ is equal to $d_{k}$, then the jth sample is labeled as the kth level. The clustering centers are obtained in a data-driven manner on the basis of the similarity between the data, and risk classification is conducted based on the distance of the samples from each clustering center, reducing the subjectivity of the classification. After risk assessment and classification are performed via a data-driven approach, risk prediction models need to be established to identify the hazards of heavy metals in grain processing products at an early stage and consequently address them before they become real risks.

### 3.3. Voting-Based Ensemble Deep Learning Method for Multi-Step Prediction

#### 3.3.1. Multi-Step Prediction

After risk assessment and classification are performed via a data-driven approach, we select the multi-step prediction method to forecast the risk level based on the past data.

**Definition** **1.**
*(*

τ

*-step prediction) Given a set of*

N

*time series*

$D={\left\{x_{i},y_{i}\right\}}_{i=1}^{N}$

*with*

xi=(xi1,⋯,xin)∈ℝp×n

*denoting that the instance*

$x_{i}$

*has the length*

n

*, dimension*

p

*, and*

d

*labels as*

$y_{i}=\left(y_{i}^{1},\cdots ,y_{i}^{d}\right)$

*to predict the future*

τ

*-step*

y˜i=(y˜i1,⋯,y˜iτ)∈ℝd×τ

*with*

τ>0

*. Figure 4 presents a schematic of a 7-step prediction.*


That is, in the proposed prediction method, the model input is given by the values of the previous *n* days, and the output is the predicted value of the subsequent day. By constantly adding the value of the prediction day, the value of the next *n* days can be predicted. However, if the predicted number of days exceeds *n* days (*n*-step), the model input no longer contains the true value. Thus, the sequence length used to evaluate the model in this study is the same as the selected step size.

#### 3.3.2. Deep Neural Network Model

RNNs have gained popularity in deep learning because of their ability to overcome the limitation of shallow neural network architectures in learning long sequences [48]. LSTM models and their variant GRU based on RNN have been built for a time series prediction. Owing to its distinct gate structure, the LSTM neural network is highly suitable for processing time series data [49]. A simplified version of LSTM, the GRU combines the forget and input gates to form an “update” gate. Thus, it has fewer parameters but less complexity, compared with LSTM. The internal structures of the RNN, LSTM, and GRU neuron modules are shown in Figure 5.

#### 3.3.3. Voting-Ensemble Method

The voting-ensemble method is primarily aimed at overcoming the limitations of various algorithms [29]. For instance, LSTM handles information at different distances from time points in various patterns. By contrast, RNNs treat each time point equally, and the GRU is located between the points; the information at certain distances from certain time points that deserves emphasis is unknown. Thus, the voting-ensemble method is applied for integrated prediction. The integrated workflow of the voting-ensemble method is shown in Figure 6.

The voting-ensemble method is used to separately integrate multiple sub-models; the obtained sub-models are arranged and combined by voting integration, which combines the final prediction results of sub-models. On the basis of this technique, the sub-model prediction results are statistically compared and analyzed, and the model with the highest prediction accuracy and overall balance is selected.

We use $\zeta_{i}$ to represent the sub-model i in the voting-ensemble algorithm; K is the K-means++ algorithm; $w_{i}$ denotes the weight assigned to each sub-model, and the sum of the weights of all sub-models should be 1; X denotes the assessment indexes. Thus, the output $h_{t}$ of the voting-ensemble method under certain days t is given by
(14)$$\begin{array}{c} h_{t}=\sum_{i}w_{i}\zeta_{i}\left(X\right)\\ \sum_{i}w_{i}=1 \end{array}$$

Thus, through the K-means++ algorithm, the final risk level output is determined as $K\left(h_{t}\right)$. This study uses the following function to indicate whether the model correctly predicts the risk level:(15)$$I\left(K\left(h_{t}\right)=K\left(y_{t}\right)\right)=\{\begin{array}{l} 1,K\left(h_{t}\right)=K\left(y_{t}\right)\\ 0,K\left(h_{t}\right)\neq K\left(y_{t}\right) \end{array}$$
where $K\left(y_{t}\right)$ is the real risk level. Therefore, the final output model meets the following requirements:(16)$$H\left(X\right)=\underset{w}{\arg\max}\sum_{t=1}^{n}I\left(K\left(h_{t}\right)=K\left(y_{t}\right)\right)$$

That is, the optimal prediction model is given.

Finally, with GRU, LSTM, and RNN as sub-models, the overall architecture of the proposed voting ensemble method integrated with deep learning models is used to calculate NIPI, as shown in Figure 7.

With the assessment index time series as the input of the voting-ensemble method, the predicted risk assessment index as the output is obtained. This output, combined with the clustering algorithm, can result in a risk level prediction.

## 4. Experiments and Results

### 4.1. Model Evaluation Index

#### 4.1.1. Prediction Performance Evaluation Index

To evaluate the prediction efficiency of the proposed multi-step food safety risk assessment index prediction method, we use the root mean square error (RMSE) and the mean absolute error (MAE). These two indicators are calculated as follows:(17)$$\mathrm{RMSE}=\sqrt{\frac{1}{n}\sum_{i=1}^{n}{\left(x_{i}-{\hat{x}}_{i}\right)}^{2}}$$
(18)$$\mathrm{MAE}=\frac{1}{n}\sum_{i=1}^{n}\left|x_{i}-{\hat{x}}_{i}\right|$$
where $x_{i}$ represents the actual value of the assessment indexes at day i, and ${\hat{x}}_{i}$ is the predicted value.

However, the prediction of the final risk level is influenced by the combination of the three assessment indexes; thus, the performance of the single assessment index, as well as the accuracy of the final prediction risk level determined by the three indexes, needs to be assessed.

#### 4.1.2. Prediction Accuracy Evaluation Index

We use the correct rate of predicting the risk level to measure the accuracy of the model, thereby predicting the risk level. When the food safety risk level as the model output is the same as the actual food safety risk level, the food safety risk level is recorded as 1; otherwise, it is 0. The level of predictive accuracy is thus calculated as follows:(19)$$PA=\frac{\sum_{t}I\left(K\left(h_{t}\right)=K\left(y_{t}\right)\right)}{t}$$
where t denotes the number of days predicted.

### 4.2. Risk Assessment and Classification

#### 4.2.1. Comprehensive Assessment Indexes

To comprehensively evaluate the heavy metal hazard in grain processing products, we first calculate the assessment indexes and used the AHP-EW method to reduce the dimensionality of the data. The EW weights of NIPI, TCR, and THQ in the AHP-EW method for three heavy metals are listed in Table 3.

The final sets of comprehensive heavy metal indexes from March to December 2020 are shown in Figure 8.

Figure 8 presents the assessment indexes exhibiting similar high-dimensional attributes, complexity, discreteness, and nonlinearity, compared with the inspection data. Therefore, the deep learning method is more suitable for assessment index prediction.

#### 4.2.2. Determination of the Risk Level

After the assessment indexes are determined, the characteristics of food safety data become a complex nonlinear time series, including abnormal data. Therefore, data normalization is necessary [14]. In the current study, NIPI, TCR, and THQ are selected as features based on K-means++ clustering in a data-driven manner. Table 4 lists the scores of clustering categories from 3 to 7 through the silhouette coefficient.

As listed in Table 4, the silhouette coefficient of Category 4 is the largest, indicating a maximum improvement in the clustering effect, which also allows risk management to perform targeted risk supervision and control. Therefore, the normalized dataset is divided into 4 categories by using the K-means++ algorithm, and the results of corresponding risk factors (i.e., NIPI, TCR, and THQ) values of each cluster center are listed in Table 5. Additionally, the risk level was determined based on the Euclidean distance between each cluster center and the origin, with a longer distance indicating a higher integrated risk.

K-means++ clustering results for the risk level are shown in Figure 9.

Subsequently, by using the K-means++ clustering algorithm to select NIPI, TCR, and THQ as the risk characteristics, this study can determine the risk level for each day and the cluster center of each risk level. In the following text, future food safety risk assessment indexes will be classified into specific risk levels based on clustering centers.

#### 4.2.3. Analysis of Heavy Metal Hazard

The risk values of the detection samples from March 2020 to December 2020 are analyzed to illustrate the advantage of risk classification based on clustering. The distribution of the risk level and clustering center is presented in Figure 10.

As shown in Figure 10, the low- and medium-risk levels comprise 87.91% of the total, and the second-highest and high-risk levels comprise 12.09% for 2020. The TCR is higher for the second-highest risk level, and the THQ is higher for the high-risk level, indicating the carcinogenic and non-carcinogenic risks in heavy metals, respectively. Moreover, in the second-highest risk and high-risk levels, three assessment indexes are higher than those of the low- and medium-risk levels. Therefore, we use the second-highest and high-risk levels as early-warning thresholds.

Considering the weekly report requirement, we perform a risk assessment of the detection results from September 9 to October 6 (Figure 11 and Figure 12) to illustrate the clustering process, combined with the assessment indexes method, to determine the risk level.

This study proposes a dynamic threshold classification method for determining the objective risk level for each day by calculating and comparing the distances (similarity) of the assessment indexes between each day and each clustering center and then selecting the class with the smallest distance as the risk classification result (Figure 11). For instance, the risk level on September 26 is assessed as high-risk because the distance is shorter to the high-risk clustering center than to other centers. We can also identify to a certain extent the causes of different risk levels through assessment indexes (Figure 12).

As shown in Figure 12, most samples are low- and medium-risk. However, the risk level corresponding to 3 October is the second-highest, which is mainly attributable to the high TCR. The highest risk level recorded, which corresponds to 17 September, is mainly attributed to the high TCR and THQ, with TCR contributing more. The high-risk classification for 26 September is caused by THQ exceeding the mean high risk. Therefore, targeted policies in risk management can be implemented to tackle different situations. The establishment of the risk classification model can identify the heavy metal hazards and interpret the factors underlying the risks. To resolve these hazards before they develop into real risks, we establish a risk level prediction model.

### 4.3. Determination of the Sub-Model

#### 4.3.1. Dataset Division and Implementation Environments

To evaluate the performance and generalizing capability of the proposed method, we select three datasets from different time periods, and each dataset is divided into training and test sets. In Dataset1 and Dataset2, we select the same dataset split ratio with different test sets to verify the generalizing capability of the model on different datasets; in Dataset2 and Dataset3, we used the same test set with different training sets and test set ratios to verify the effect of the data split ratio on the model (Figure 13). Their generalizing capabilities and performance in three datasets are ultimately measured.

In this paper, we deploy deep learning models like RNN, GRU, and LSTM with Tensorflow 2.0.0 using the Keras package. All the models were programmed by Python 3.6 and trained on a laptop computer (Intel i5-1035G1 CPU, without GPU used as the data accelerator).

#### 4.3.2. Performance of the Sub-Model

Food regulatory agencies require weekly detection reports. Considering this requirement, together with the limitations of the multi-step time series prediction of long-term error accumulation, this study chooses a time step of 21 to compare the sub-models, which include several existing typical machine learning or deep learning models, focusing primarily on the predictive efficiency of the 7-day model. The 14- and 21-day models are used for an auxiliary comparison via MAE and RMSE indicators. The predictive accuracy rates of the sub-models are also compared.

To improve the comparison of the prediction performances of the different models and select the appropriate sub-models, we visualize the RMSE and MAE of sub-models for the 7, 14, and 21-day prediction results on a heatmap (Figure 14).

As shown in Figure 14, as the number of prediction steps increases, the MAE and RMSE values of each model increase as well. The reason is that the larger the number of prediction steps, the more information is missing and the lower the prediction accuracy. Notably, the following instance is also observed: in the NIPI of Dataset2, the 7-day MAE of the RNN model is 0.1263, and the 14-day MAE is 0.0814. The reason is that in the multi-step prediction, positive and negative errors are offset as errors accumulate, hence the decrease in RMSE and MAE values over time. Figure 14 also shows that RNN, GRU, and LSTM perform better than the other models, but the parts of the models only slightly vary. Therefore, we determine the final sub-model portfolio on the basis of the accuracy of the risk level prediction.

#### 4.3.3. Comparison of Different Sub-Models

The correct accuracy rate of each sub-model in risk level prediction is shown in Figure 15.

In Figure 15, the effects of eXtreme Gradient Boosting (XGBoost) and BP models are worse than those of the other models, and the accuracy rates in the subsequent 7-, 14-, and 21-day periods are lower than the average; by contrast, the GRU model performs more efficiently in the subsequent 7-, 14-, and 21-day periods. The predictive efficiency of the RNN model is poor in the subsequent 7-day period; however, this characteristic improves in the 14- and 21-day periods, potentially reaching a relatively satisfactory level. When LSTM predicts the subsequent 14- and 21-day periods, the predictive efficiency is low, but when it predicts the subsequent 7-day period, the predictive efficiency is high, reaching 85.71%. Although the AHC-RBF model outperforms LSTM in predictive accuracy in the 14-day period, a large discrepancy in the predictive efficiency of LSTM is observed in the 7-day period, which is more important than the 14- and 21-day because those periods have the presence of cumulative errors and the requirement of a weekly report. Meanwhile, the proposed approach requires that the sub-models exhibit satisfactory and similar performances. Therefore, the combination of the RNN, GRU, and LSTM models is superior to other models, and these three models possess different accuracy attributes at different prediction steps. This study then integrates the three models into the voting-ensemble method discussed in the subsequent sub-section.

### 4.4. Voting-Ensemble Model

#### 4.4.1. Performance of the Voting-Ensemble Model

To verify the efficiency of the proposed model, we compare sub-models and existing models by using the proposed voting-ensemble method on the same detection data. The prediction performance of the sub-models and the voting-ensemble method is summarized in Table 6.

As shown in Table 6, in these three datasets, the proposed voting-ensemble method has the smallest RMSE and MAE values on almost every indicator, performing better than the sub-models. Similarly, on different datasets, the predictive efficiency of each model declines as the number of prediction steps increases.

In terms of the time required for the training process, as illustrated in Table 7, RNN and GRU are significantly faster; they are about two times faster than the LSTM, and about four times faster than the proposed voting-ensemble model due to its requirement of waiting until the end of the training of the sub-models. Thus, when the proposed voting-ensemble method is used to make an early-warning analysis of food safety, the training time is extended.

#### 4.4.2. Comparison with Sub-Models

The prediction and actual curves generated using the RNN, GRU, LSTM, and the proposed method in each dataset are presented in Figure 16, Figure 17 and Figure 18.

As shown in Figure 16, Figure 17 and Figure 18, the voting-ensemble model has a higher degree of coincidence between the actual value curve and the predicted value curve on the test set than those of the sub-models. This result is similar to the outcome summarized in Table 6, indicating that the voting-ensemble model exhibits powerful performance and predictive capabilities.

In Figure 19, the proposed voting-ensemble model exhibits the highest accuracy. This finding, combined with the results in Figure 15, indicates that the accuracy rates of the voting-ensemble method for 7-, 14-, and 21-day periods exceed those of each sub-model. The average predictive accuracy of the three datasets in the 7-day period reaches 90.47%, which is higher than those of the sub-models and existing models. According to the results in Table 6, the RMSE and MAE of the proposed method are less than the values obtained using other methods. Thus, compared with other methods, the proposed method has a better predictive performance.

## 5. Discussion

In this study, we established a novel time series multi-step prediction model for classifying and assessing the risk levels of heavy metal hazards in grain processing products. Food safety assessment indexes were introduced to explain the heavy metal hazards. The data-driven clustering algorithm reduced the subjectivity of threshold determination. We then introduced the deep learning method in early-warning systems in the food industry to implement a multi-step time series prediction and validate its efficiency by comparing it with existing models.

### 5.1. Risk Assessment and Classification

Recent studies have focused on conducting a risk assessment of food contaminants, in addition to a food safety risk evaluation based on the calculation results for the detection samples via AHP-EW, in fields implementing early-warning systems. The traditional approach is based on risk values for establishing early-warning models, which lack the systematic measurement of food contaminant hazards [14]. Alternatively, we established a risk assessment model by using the NIPI, TCR, and THQ to satisfy the comprehensive evaluation requirements of risk management to a certain extent. Therefore, this study realized a systematic dietary analysis of food safety risks by introducing assessment indexes (Figure 8). Meanwhile, to reduce regulatory costs, risk levels need to be assessed and different risk levels have to be prioritized differently. However, risk level classification by setting absolute thresholds is subjective [19].

Therefore, regarding the risk classification and analysis of heavy metal hazards, an assessment index-based risk classification by cluster analysis uses silhouette coefficients to determine optimal and risk level classification (Table 4) and obtain clustering results (Figure 9 and Figure 10). With this approach, we can determine the relative threshold for comprehensive indexes in a data-driven manner and objectively analyze heavy metal hazards (Figure 11). We can also identify the causes of each risk level and evaluate the effect of each index on the classification so that risk management can achieve a retrospective analysis of food safety risks and develop targeted strategies (Figure 12).

Compared with existing risk assessment and classification methods enabled by early-warning models, the proposed risk level framework in this study provides an interpretable risk assessment, in addition to data-driven and objective risk classification based on a dietary exposure assessment and K-means++ clustering algorithms. It can be used by risk management departments in assessing the comprehensive relative risk of heavy metal hazards and determining risk levels. With this tool, measures and policies may be implemented to address and retrace the factors that contribute to different risk levels for efficient food safety risk management.

### 5.2. Multi-Step Time Series Prediction of Risk Levels

In fields using food safety early-warning systems, most studies use single-step risk prediction. In the current study, we employed multi-step prediction, as opposed to single-step prediction, for the assessment index time series, which can predict data that have not occurred. However, a multi-step or long-term prediction is difficult and challenging due to the lack of information and uncertainty [50] or error accumulation (Figure 14, Table 6). Therefore, models with a satisfactory performance need to be developed to improve the accuracy of a multi-step prediction.

Research on time series predictions in the food early-warning field using the machine learning method or ANNs because of the nonlinear characteristics (Figure 2 and Figure 8) of food safety time series in practice is required [12]. Compared with traditional ANNs, deep learning methods such as RNN, GRU, and LSTM, can capture long-term time series data [33]; this finding is consistent with the performance of the sub-models (Figure 14). To further improve the accuracy of a multi-step prediction, we proposed the voting-ensemble method to integrate the advantages of different models and select the sub-models with satisfactory performances (Figure 15) to establish the voting-ensemble based deep learning method [29]. The final result suggests that the accuracy of the proposed method reaches 90.47% in 7 days, which meets the weekly report requirement set by the risk management department.

### 5.3. Limitations

The voting-ensemble method was selected in this study to integrate various deep learning models and thereby achieve an improved accuracy rate; however, the training time is extended (Table 7). Despite the high time complexity of the proposed model, it is not unacceptable, as computing power continues to increase. Meanwhile, the outcomes of data grading are acquired in a data-driven way; the greater the amount of data collected, the higher the accuracy of risk grading. Therefore, we will continue to track food safety data to obtain risk classification results with increased accuracy.

## 6. Conclusions

To establish an early-warning model that can systematically assess the risk of heavy metals in grain processing products, this study proposed a novel multi-step time series prediction model based on a deep learning method. By adopting a voting-ensemble method, this study increased the accuracy of the prediction model. The final results also indicate that the proposed model achieves an accuracy of 90.47%, which meets the weekly food sampling report requirement for risk management. Meanwhile, risk classification based on system assessment allows food regulatory authorities to objectively prioritize and identify the causes of risk, thus enhancing the early control of food safety risks and reducing the costs of risk management. Moreover, an early-warning system based on deep learning models in a multi-step time series prediction, instead of the existing single-step or fitting prediction machine learning model, can more efficiently capture the dynamic operation pattern of a food safety time series. It can also further enable operators to detect food safety risks promptly, as well as improve early-warning systems for food safety, allowing for a continuous and interactive process to address future problems [51]. The food safety supervision departments can strengthen the supervision of heavy metal hazards based on the proposed early-warning model.

In future research, we intend to track food safety data to obtain risk classification results with increased accuracy and attempt different voting approaches to achieve enhanced multi-step prediction accuracy.

## Figures and Tables

**Figure 1 foods-11-00823-f001:**
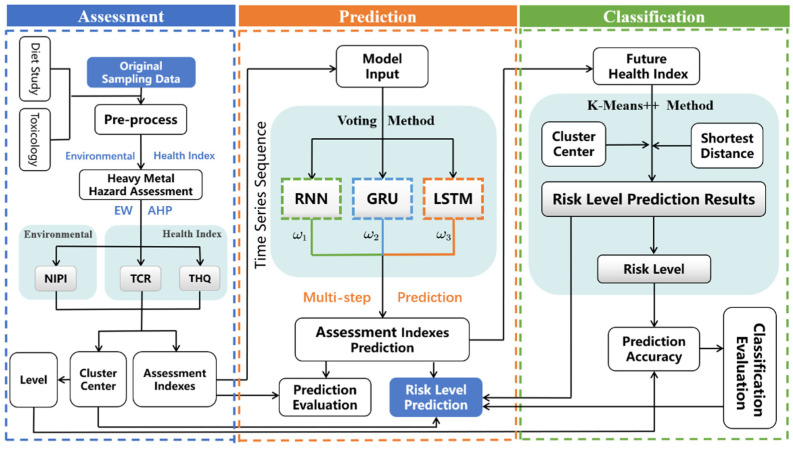
Framework of the classification and prediction of food safety risk level.

**Figure 2 foods-11-00823-f002:**
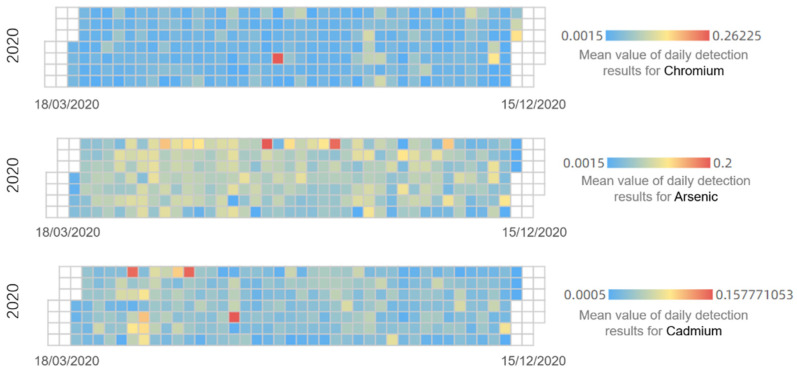
Mean values of daily detection results for chromium, arsenic, and cadmium.

**Figure 3 foods-11-00823-f003:**
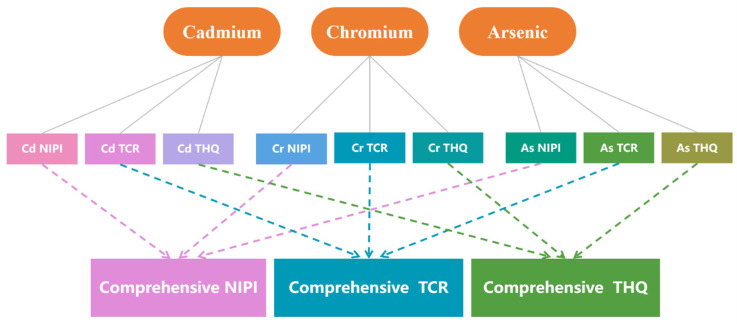
Calculation of the risk assessment index using the AHP-EW.

**Figure 4 foods-11-00823-f004:**
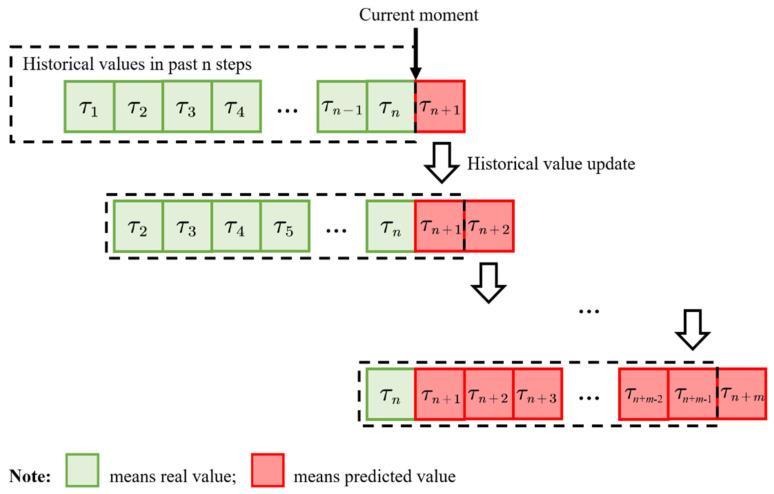
Schematic of multi-step prediction (7-step, *n* = m = 7; satisfy the weekly report requirement of food supervision department).

**Figure 5 foods-11-00823-f005:**
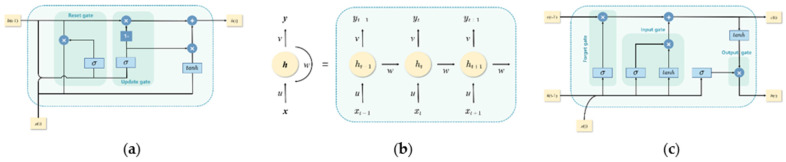
GRU, RNN, and LSTM neuron modules; (**a**) gated recurrent units (GRUs), (**b**) recurrent neural networks (RNN), (**c**) long short-term memory (LSTM); where $h_{t-1}$ and $h_{t}$ represent the outputs of the module at time t−1 and t, respectively; $x_{t}$ and $y_{t}$ denote the input and output of the module at time t; $c_{t}$ indicates the state information of the module at time t; and σ and tanh are the sigmoid and hyperbolic tangent activation functions.

**Figure 6 foods-11-00823-f006:**
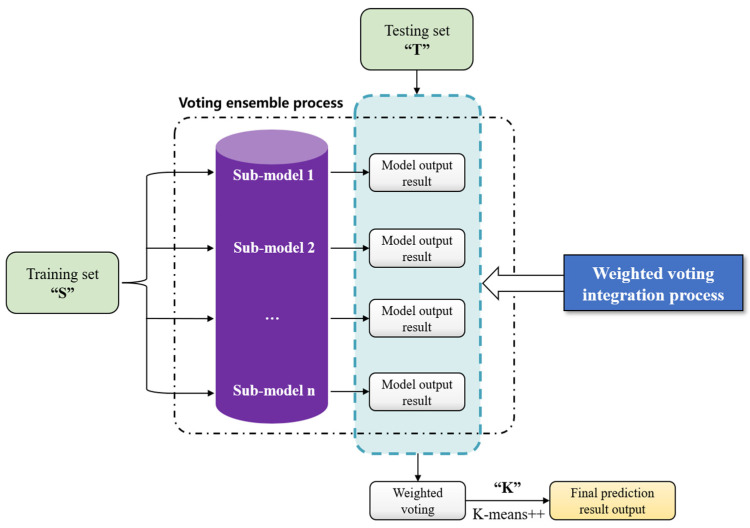
Schematic of the voting-ensemble method.

**Figure 7 foods-11-00823-f007:**
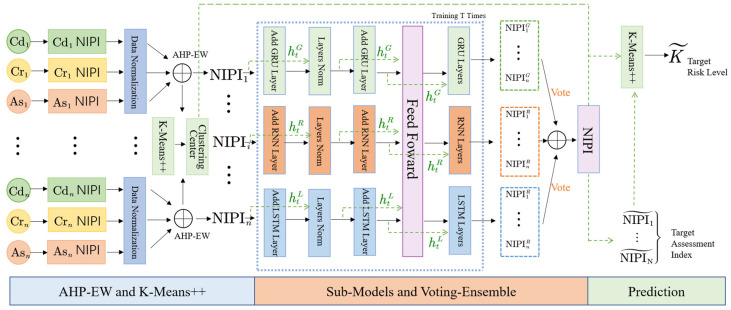
Overall architecture of the proposed voting-ensemble method (NIPI).

**Figure 8 foods-11-00823-f008:**
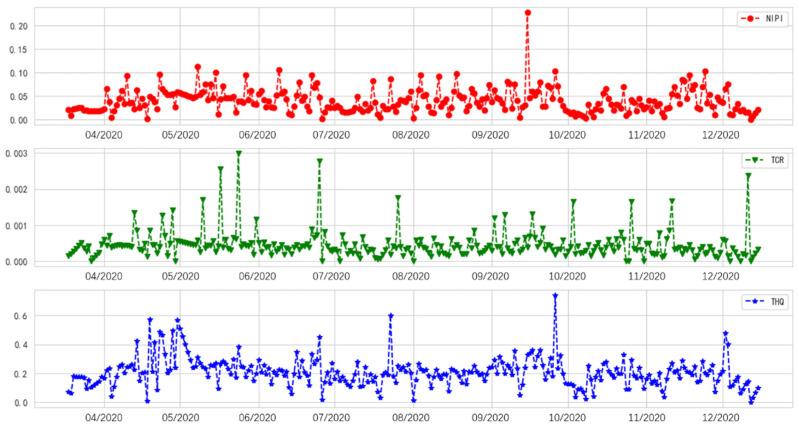
Assessment index calculation using the AHP-EW.

**Figure 9 foods-11-00823-f009:**
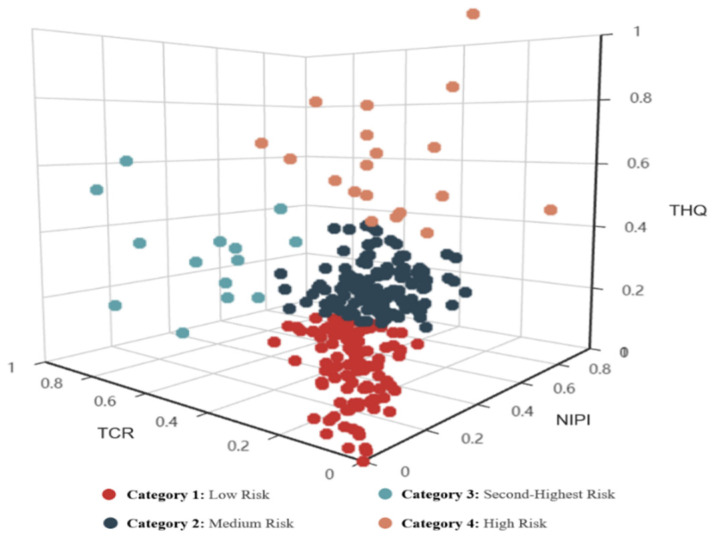
K-means++ clustering results.

**Figure 10 foods-11-00823-f010:**
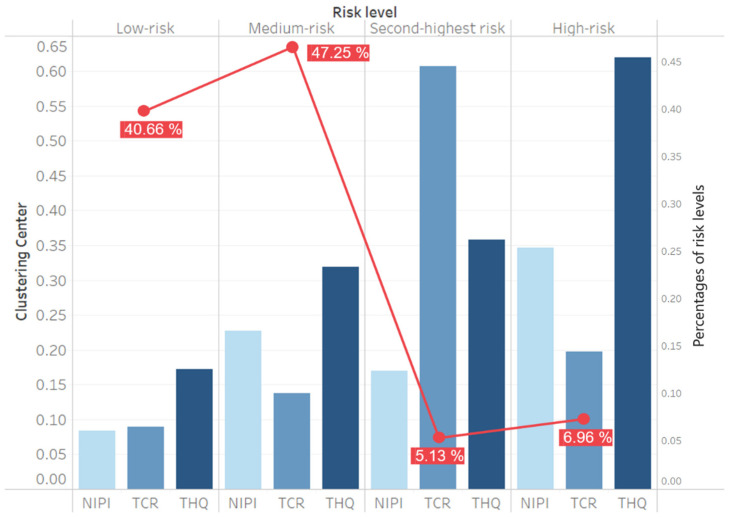
Distribution of different risk levels.

**Figure 11 foods-11-00823-f011:**
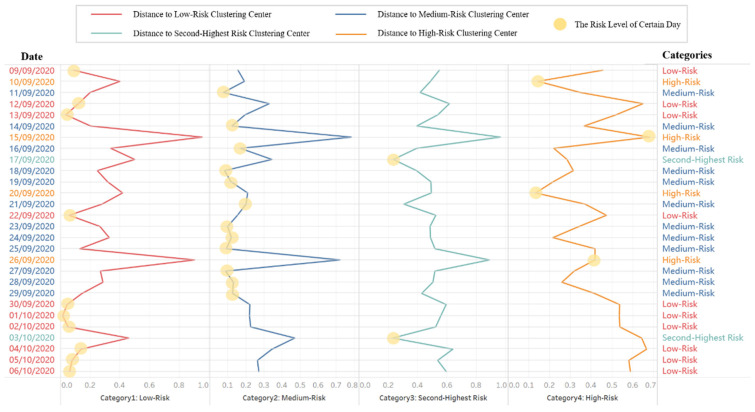
Threshold of risk classification determined by the distance to the clustering center.

**Figure 12 foods-11-00823-f012:**
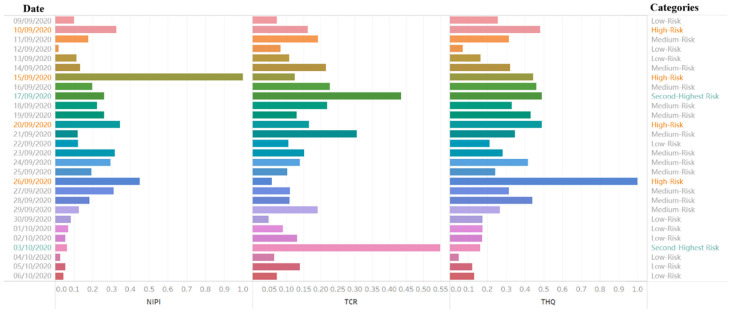
Risk levels and assessment indexes for certain days.

**Figure 13 foods-11-00823-f013:**
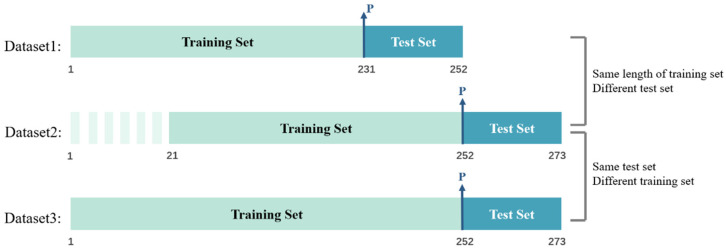
Diagram of the three dataset divisions.

**Figure 14 foods-11-00823-f014:**
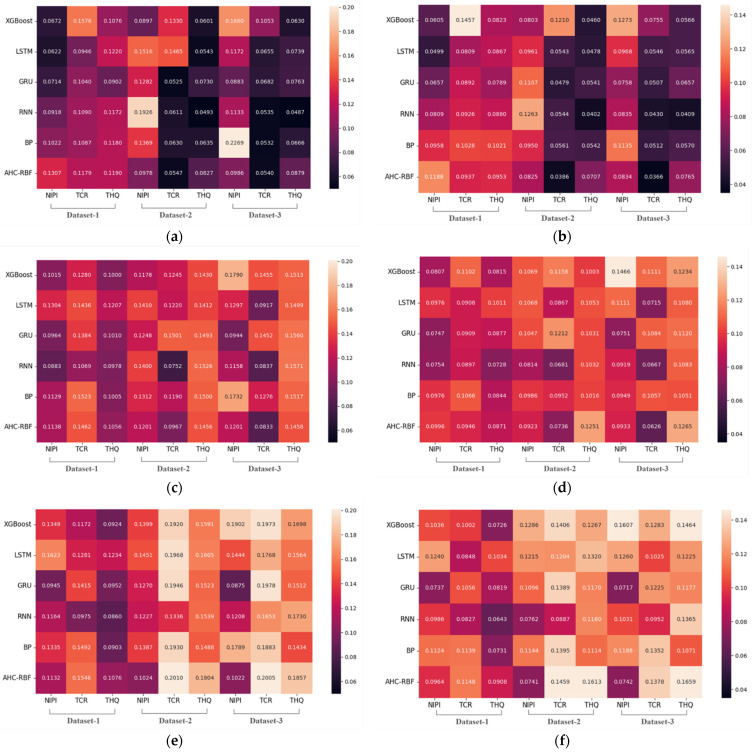
RMSE and MAE of the sub-models for the 7-, 14-, and 21-day predictions. (**a**) RMSE (7 days), (**b**) MAE (7 days), (**c**) RMSE (14 days), (**d**) MAE (14 days), (**e**) RMSE (21 days), (**f**) MAE (21 days).

**Figure 15 foods-11-00823-f015:**
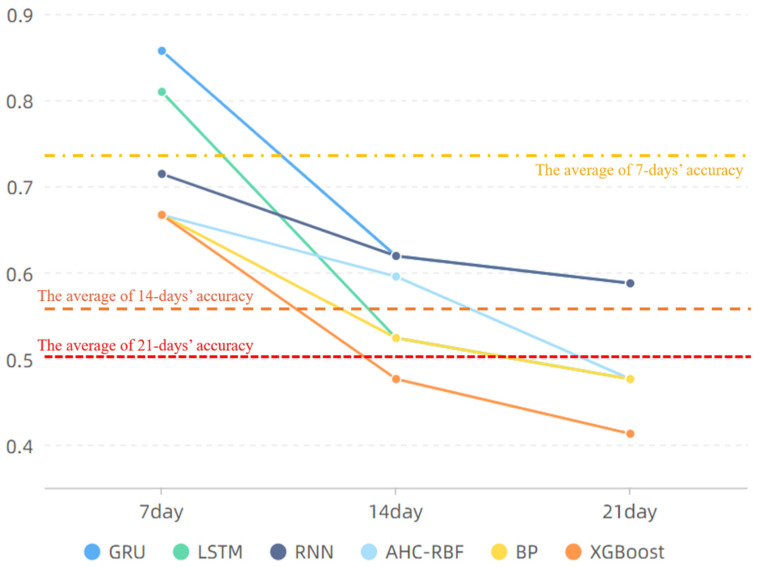
Correct rate of each sub-model.

**Figure 16 foods-11-00823-f016:**
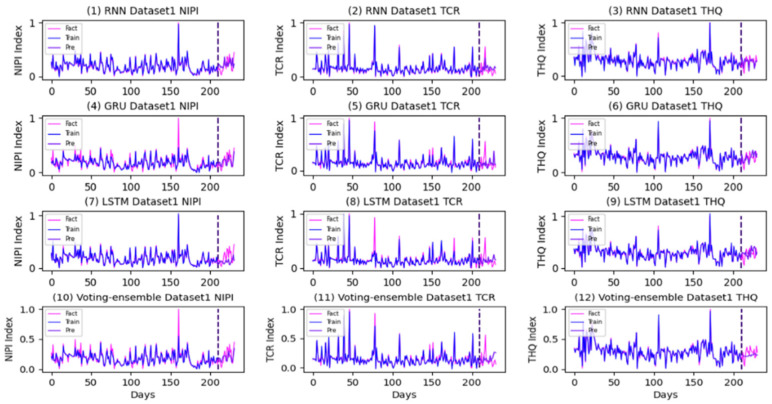
Prediction and actual curves generated using the RNN, GRU, LSTM, and the proposed method in Dataset1.

**Figure 17 foods-11-00823-f017:**
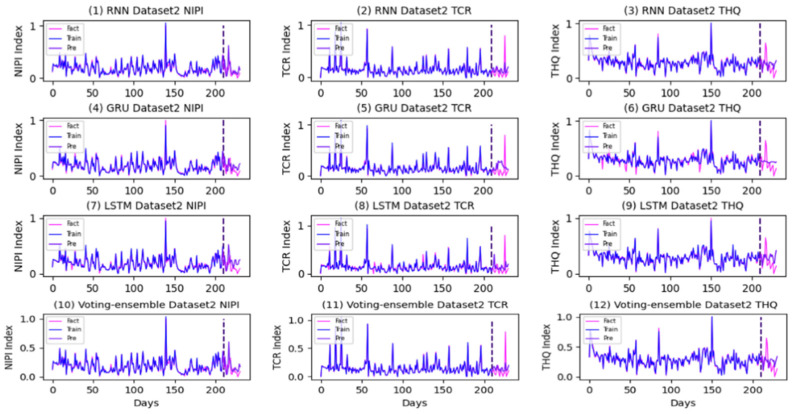
Prediction and actual curves generated using the RNN, GRU, LSTM, and the proposed method in Dataset2.

**Figure 18 foods-11-00823-f018:**
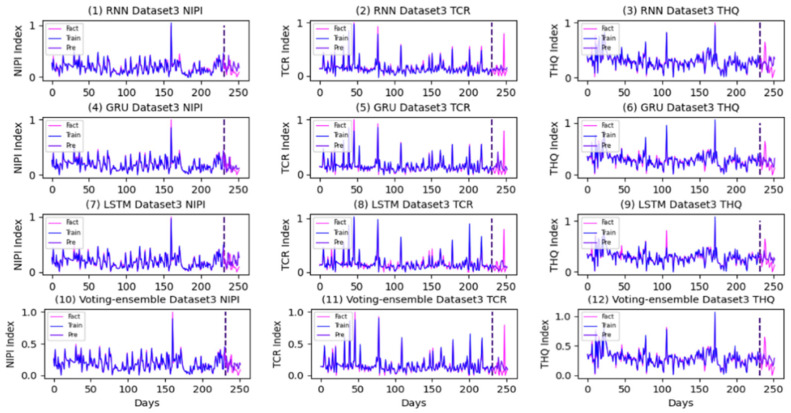
Prediction and actual curves generated using the RNN, GRU, LSTM, and the proposed method in Dataset3.

**Figure 19 foods-11-00823-f019:**
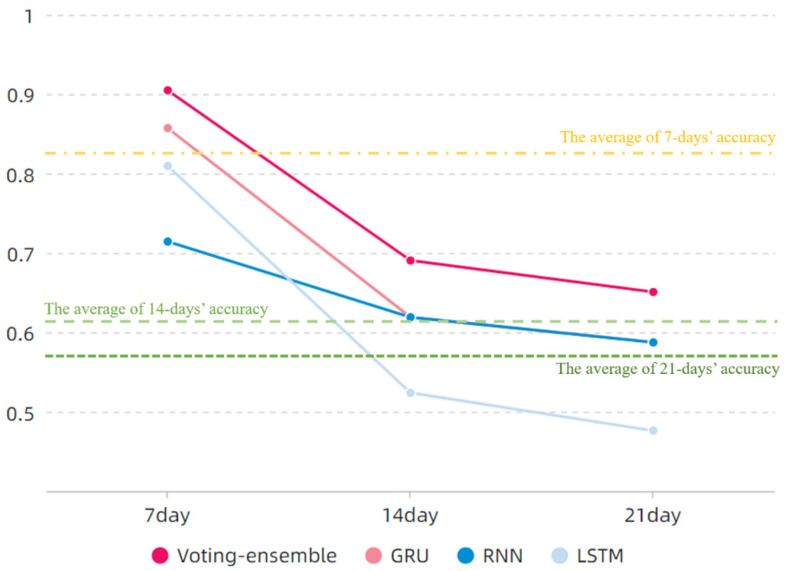
Correct rates of RNN, GRU, LSTM, and the proposed method.

**Table 1 foods-11-00823-t001:** A description of the several observations in the raw data set.

Heavy Metal Elements	Date of Inspection	Unit	Food Category	Commodity Name	Province	Inspection Result	Inspection Standard
Inorganic arsenic(As)	6/23/2020	mg/kg	Grain processing products	Shrimp rice on Southern Margin	Hunan	0.08	GB 5009.11-2014
Cadmium(Cd)	9/9/2020	mg/kg	Grain processing products	Mojiang purple rice	Henan	0.071	GB 5009.15-2014
Inorganic arsenic(As)	6/24/2020	mg/kg	Grain processing products	Organic rice	Zhejiang	Not detected	GB 5009.11-2014
Cadmium(Cd)	7/29/2020	mg/kg	Grain processing products	Tatai Oil viscose rice	Guangdong	0.23	GB 5009.15-2014
Cadmium(Cd)	8/11/2020	mg/kg	Grain processing products	Superior fragrant sticky rice	Hainan	0.37	GB 5009.15-2014
Chromium(Cr)	8/27/2020	mg/kg	Grain processing products	Water-milled glutinous rice flour	Shanxi	Not detected	GB 5009.123-2014

**Table 2 foods-11-00823-t002:** Grain processing product consumption (g/day) of residents.

**Province**	**Heilongjiang **	**Jilin **	**Liaoning **	**Beijing **	**Hebei **	**Henan **	**Ningxia **	**Shaanxi **	**Inner ** **Mongolia **	**Qinghai **
Grain processing products	673.70	1201.02	1131.27	825.20	935.58	1517.90	1002.35	783.86	1038.39	1681.60
**Province**	**Fujian **	**Zhejiang **	**Jiangsu **	**Shanghai **	**Jiangxi **	**Hubei **	**Sichuan **	**Hunan **	**Guangxi **	**Guangdong **
Grain processing products	920.55	1126.50	620.64	566.96	641.55	916.16	806.08	905.68	765.55	431.90

**Table 3 foods-11-00823-t003:** Entropy weights of NIPI, TCR, and THQ for chromium, cadmium, and arsenic.

Heavy Metal	Chromium			Cadmium			Arsenic		
Assessment Index	NIPI	TCR	THQ	NIPI	TCR	THQ	NIPI	TCR	THQ
EW-Weights	0.1285	0.2525	0.1215	0.092	0.1999	0.0688	0.021	0.0929	0.0228

**Table 4 foods-11-00823-t004:** Silhouette coefficients of different categories.

Categories	3	4	5	6	7
Silhouette coefficient	0.37741	0.38230	0.36543	0.36067	0.30792

**Table 5 foods-11-00823-t005:** Clustering center of the assessment indexes (normalized); based on the Euclidean distance between each cluster center and the origin, the cluster centers are marked as different risk levels.

Clustering Center	NIPI	TCR	THQ	Distance to Origin	Risk Level
Category 1	0.0841556	0.0898703	0.1717660	0.21133483	Low-Risk
Category 2	0.2272796	0.1375510	0.3194492	0.41548066	Medium-Risk
Category 3	0.1693235	0.6068999	0.3582518	0.72480507	Second-Highest Risk
Category 4	0.3463198	0.1970221	0.6196991	0.73673754	High-Risk

**Table 6 foods-11-00823-t006:** Performance of LSTM, GRU, RNN, and the voting-ensemble method (21 steps).

Model	Evaluation	Days	Dataset1			Dataset2			Dataset3		
	Indicator		NIPI	TCR	THQ	NIPI	TCR	THQ	NIPI	TCR	THQ
LSTM	RMSE	7	0.0622	0.0946	0.1220	0.1516	0.1465	0.0543	0.1172	0.0655	0.0739
		14	0.1304	0.1436	0.1207	0.1410	0.122	0.1412	0.1297	0.0917	0.1499
		21	0.1623	0.1281	0.1234	0.1451	0.1968	0.1605	0.1444	0.1768	0.1564
		Avg	0.1183	0.1221	0.1220	0.1459	0.1551	0.1187	0.1304	0.1113	0.1267
	MAE	7	0.0499	0.0809	0.0867	0.0961	0.0543	0.0478	0.0968	0.0546	0.0565
		14	0.0976	0.0908	0.1011	0.1068	0.0867	0.1053	0.1111	0.0715	0.1080
		21	0.1240	0.0848	0.1034	0.1215	0.1204	0.1320	0.1260	0.1025	0.1225
		Avg	0.0905	0.0855	0.0971	0.1081	0.0871	0.095	0.1113	0.0762	0.0957
GRU	RMSE	7	0.0714	0.104	0.0902	0.1282	0.0525	0.0730	0.0883	0.0682	0.0763
		14	0.0964	0.1384	0.1010	0.1248	0.1501	0.1493	0.0944	0.1452	0.156
		21	0.0945	0.1415	0.0952	0.1270	0.1946	0.1523	0.0875	0.1978	0.1512
		Avg	0.0874	0.1280	0.0955	0.1267	0.1324	0.1249	0.0901	0.1371	0.1278
	MAE	7	0.0657	0.0892	0.0789	0.1107	0.0479	0.0541	0.0758	0.0507	0.0657
		14	0.0747	0.0909	0.0877	0.1047	0.1212	0.1031	0.0751	0.1084	0.1120
		21	0.0737	0.1056	0.0819	0.1096	0.1389	0.1170	0.0717	0.1225	0.1177
		Avg	0.0714	0.0952	0.0828	0.1083	0.1027	0.0914	0.0742	0.0939	0.0985
RNN	RMSE	7	0.0918	0.109	0.1172	0.1926	0.0611	0.0493	0.1133	0.0535	0.0487
		14	0.0883	0.1069	0.0978	0.1400	0.0752	0.1526	0.1158	0.0837	0.1571
		21	0.1164	0.0975	0.086	0.1227	0.1336	0.1539	0.1208	0.1653	0.1730
		Avg	0.0988	0.1045	0.1003	0.1518	0.089	0.1186	0.1166	0.1008	0.1263
	MAE	7	0.0809	0.0926	0.088	0.1263	0.0544	0.0402	0.0835	0.043	0.0409
		14	0.0754	0.0897	0.0728	0.0814	0.0681	0.1032	0.0919	0.0667	0.1083
		21	0.0986	0.0827	0.0643	0.0762	0.0887	0.118	0.1031	0.0952	0.1365
		Avg	0.0850	0.0883	0.0750	0.0946	0.0704	0.0871	0.0928	0.0683	0.0952
Voting-ensemble	RMSE	7	0.0663	0.1028	0.0908	0.1852	0.0683	0.0498	0.0621	0.0625	0.0662
		14	0.0944	0.1382	0.0998	0.1370	0.0772	0.1496	0.0835	0.1144	0.1510
		21	0.0978	0.1394	0.0936	0.1229	0.1420	0.1538	0.0894	0.1838	0.1495
		Avg	0.0862	0.1268	0.0947	0.1484	0.0958	0.1177	0.0783	0.1202	0.1222
	MAE	7	0.0618	0.0884	0.078	0.1205	0.0577	0.0413	0.0479	0.0518	0.0586
		14	0.0731	0.0908	0.0863	0.0822	0.0686	0.1015	0.0665	0.0939	0.1074
		21	0.0774	0.1036	0.0803	0.0818	0.0925	0.1191	0.0769	0.1145	0.1173
		Avg	0.0708	0.0943	0.0815	0.0948	0.0729	0.0873	0.0638	0.0867	0.0944

**Table 7 foods-11-00823-t007:** The running time of the sub-models and the proposed voting-ensemble model.

Model	RNN	GRU	LSTM	Voting-Ensemble
Running Time/s	441	551	1043	2145

## Data Availability

Restrictions apply to the availability of these data. Data was obtained from the State Administration for Market Regulation statistics and are available at http://spcj.gsxt.gov.cn with the permission of the State Administration for Market Regulation statistics.

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
