# Peer review of "A Voting-Based Ensemble Deep Learning Method Focused on Multi-Step Prediction of Food Safety Risk Levels: Applications in Hazard Analysis of Heavy Metals in Grain Processing Products"

_foods, 2022, doi:10.3390/foods11060823_

Round 1

Reviewer 1 Report

Article with the title "A voting-based ensemble deep learning method focused on multistep prediction of food safety risk levels: Applications in hazard analysis of heavy metals in grain processing products" includes 24 pages of text along with 19 figures and 5 tables. It is based on 51 references, mostly not older than 5 years.

The short introduction is elaborated with the definition of the basic issues for the research described below. Quite interesting is the inclusion of the second chapter describing background studies.

In the material and methods chapter authors describe an area I do not want to comment on, I am not an expert in this field, but I could say that it contains all the essential information and it describes the generally required information for the experiment in these studies.

The material and methods describe an area that I do not want to comment on, I am not an expert in the field, yet I dare say that it contains all the essentials and it describes the generally required information for the experiment in these studies. The discussion is included in a separate chapter.

The conclusion of the study is clear and contains a commitment and a plan for future research.
I have no major comments on the text. Nevertheless, I lack information on the application of the conclusion for practical use.

There are also minor styling errors that can be corrected when editing text.

Author Response

Thanks very much for taking your valuable time to review this manuscript. We really appreciate all your generous comments and suggestions!

Reviewer 2 Report

The paper ‘A voting-based ensemble deep learning method focused on multistep prediction of food safety risk levels: Applications in hazard analysis of heavy metals in grain processing products’ is a nice trial to introduce operational research methodology in food industry. Authors try to deliver a new solution (model) for the systematic assessment of the risk level to control heavy metal hazards in grain processing products. The paper is readable and written in details being interesting for the community, but unfortunately needs some minor improvement.

Comments:

  • l. 24 – In the abstract rather the overall accuracy than the highest accuracy should be reported. The highest accuracy can be obtained by chance or special benchmarks
  • l. 144 – data should be available to confirm obtained results
  • l. 144 - The link to the produced software should be available for the reader
  • l. 144 – Detailed description of the one observation (at least) in a data set should be included
  • l. 231 – The short explanation of K-means++ algorithm is needed
  • l. 231 – the way of obtaining final number of clusters should be presented
  • l. 242 – the detailed description of distance calculation together with the definition and calculation of the cluster’s center (l. 242) should be added
  • l. 256 – the explanation of choosing 7 as a number of steps in prediction is needed
  • l. 293 – the description of statistics used for sub-model prediction results should be added
  • l. 357 – detailed justification and explanation of the building process of categories and different risk factors is needed. The definition of the ‘LowRisk’,…’High-Risk’ should be added
  • l. 403 – n-fold cross validation test is needed. Artificial once division can be misleading or explanation of the chosen strategy is needed.
  • There is a lack of information about computational efficiency of the proposed methodology. Clear description of dependency of computational time and the analyzed instances as well as description of the computational environments should be added.

Author Response

(The authors gave the same response as above.)
